# CombLM: Adapting Black-Box Language Models through Small Fine-Tuned Models

**Aitor Ormazabal**[1]     **Mikel Artetxe**[2]     **Eneko Agirre**[1]

[1]HiTZ Center, University of the Basque Country (UPV/EHU)     [2]Reka AI

aitor.ormazabal@ehu.eus     mikel@reka.ai     e.agirre@ehu.eus

## Abstract

Methods for adapting language models (LMs) to new tasks and domains have traditionally assumed white-box access to the model, and work by modifying its parameters. However, this is incompatible with a recent trend in the field, where the highest quality models are only available as black-boxes through inference APIs. Even when the model weights are available, the computational cost of fine-tuning large LMs can be prohibitive for most practitioners. In this work, we present a lightweight method for adapting large LMs to new domains and tasks, assuming no access to their weights or intermediate activations. Our approach fine-tunes a small white-box LM and combines it with the large black-box LM at the probability level through a small network, learned on a small validation set. We validate our approach by adapting a large LM (OPT-30B) to several domains and a downstream task (machine translation), observing improved performance in all cases, of up to 9%, while using a domain expert 23x smaller.

## 1 Introduction

Natural language processing (NLP) has witnessed remarkable progress in recent years thanks to the development of increasingly powerful LMs (Brown et al., 2020; Andrew and Gao, 2007; Chowdhery et al., 2022; Touvron et al., 2023). Since these models are usually generalists, it is often of interest to adapt them to new domains, underrepresented or not found in the original training data. Typically, domain adaptation techniques assume white-box access to the model parameters, for example by fine-tuning on a particular target domain (Gururangan et al., 2020).

However, this approach has become increasingly infeasible given the ongoing paradigm shift in the field—state-of-the-art models like GPT-4 and PaLM-2 are only accessible as black-boxes through inference APIs and, even when the model weights

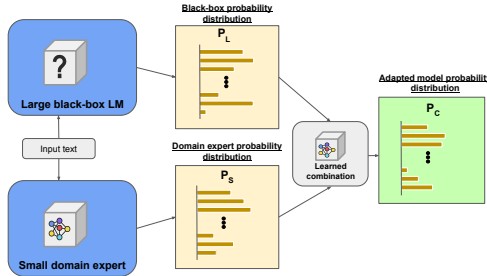

Figure 1: **Illustration of our approach.** We leverage a large black-box LM and a small white-box LM, fine-tuned on a domain-specific corpus. We combine both models' outputs at the probability level, through a combination function learned on a small fitting set, requiring very little compute. The resulting model adapts the large black-box to the target domain, performing better than either of the original ones.

are available, the computational cost of fine-tuning large models can be prohibitive. Consequently, domain adaptation methods that cannot leverage the power of black-box LLMs are likely to fall behind.

In this work, we propose a simple and lightweight approach to adapt black-box LMs to new domains, without requiring access to weights or intermediate activations. Our method consists of two main steps: (1) training a small, white-box model on the desired target domain, and (2) learning a function that combines the probability distributions from the large black-box LM and the small domain expert LM, producing a new probability distribution. The combination function is a small neural network that is trained on a small validation dataset.

We evaluate our method by adapting a black-box model to three distinct domains and a downstream task—machine translation (MT). In all cases, we observe that the combined model outperforms both the large black-box model and the small domain expert. This shows that it is possible to adapt black-box LMs to new domains, opening an exciting line

of research.

## 2 Proposed method

Our approach works in two steps: (1) we train a small domain expert LM, and (2) we learn a function that combines the outputs of the domain expert LM and a large black-box LM at the probability level.

More concretely, an LM defines a probability distribution over the possible continuations of any given text. That is, given a sequence of tokens $\mathbf{x} = (x_1, x_2, ..., x_n) \in V^*$, where $V$ is the model vocabulary, an LM parametrizes $P_{LM}(y_{next}|\mathbf{x})$, the probability that $y_{next}$ is the continuation of $\mathbf{x}$ in a text. We let $P_S$ denote our small domain expert LM, and $P_L$ denote the large black-box generalist LM. Our combination function $\mathbf{f}$ defines a new combined probability distribution $P_C$: $P_C(y_{next}|\mathbf{x}) = \mathbf{f}(P_S(\cdot|\mathbf{x}), P_L(\cdot|\mathbf{x}))_{y_{next}}$. Here $\mathbf{f} : \mathbb{R}^{|V|} \times \mathbb{R}^{|V|} \to \mathbb{R}^{|V|}$ is a vector-valued function that receives full probability distributions, and outputs a new probability distribution.

To train the domain expert LM, we fine-tune a pre-trained model on a small domain-specific dataset. For the combination function, we consider several alternatives of varying capacity:

1. **Mean.** The arithmetic mean of the two distributions: $\mathbf{f}(\mathbf{y_1}, \mathbf{y_2}) = (\mathbf{y_1} + \mathbf{y_2})/2$.

2. **Constant-scalar**. A linear combination of the two input distributions, with a constant combination factor $\lambda$: $\mathbf{f}(\mathbf{y_1}, \mathbf{y_2}) = \lambda \mathbf{y_1} + (1 - \lambda)\mathbf{y_2}$.

3. **Constant-vector**. A token-wise version of the previous combination, where $\boldsymbol{\lambda} \in \mathbb{R}^{|V|}$ is a constant vector, and the combination factor varies per-token: $\mathbf{f}(\mathbf{y_1}, \mathbf{y_2}) \propto \boldsymbol{\lambda} \circ \mathbf{y_1} + (\mathbf{1} - \boldsymbol{\lambda}) \circ \mathbf{y_2}$, where $\circ$ is the Hadamard (elementwise) product. Note the proportionality instead of equality in the definition, as a re-normalization is required when combining distributions per-token.

4. **Entropy-scalar**. A scalar $\lambda$ is predicted from the entropies of each distribution, $\lambda = g(\mathrm{H}(\mathbf{y_1}), \mathrm{H}(\mathbf{y_2}))$, and the output is a linear combination as in *constant-scalar*: $\mathbf{f}(\mathbf{y_1}, \mathbf{y_2}) = \lambda \mathbf{y_1} + (1 - \lambda)\mathbf{y_2}$. The function $g$ is parametrized by a small neural network.

5. **Entropy-vector**. An token-wise version of the previous combination, where a vector $\boldsymbol{\lambda} = $ $\mathbf{g}(\mathrm{H}(\mathbf{y_1}), \mathrm{H}(\mathbf{y_2})) \in \mathbb{R}^{|V|}$ is predicted , and then the per-token combination is done as in *constant-vector*.

6. **Full-scalar**. A single $\lambda$ is predicted from full input distributions, $\lambda = g(\mathbf{y_1}, \mathbf{y_2})$, and then the output is a linear combination as in the constant combination: $\mathbf{f}(\mathbf{y_1}, \mathbf{y_2}) = \lambda \mathbf{y_1} + (1 - \lambda)\mathbf{y_2}$. The function $g$ is parametrized by a small neural network.

7. **Full-vector**. Token-wise version of the previous combination, where a vector $\boldsymbol{\lambda} = \mathbf{g}(\mathbf{y_1}, \mathbf{y_2}) \in \mathbb{R}^{|V|}$ is predicted , and the per-token combination is done as in *constant-vector*.

On one end of the spectrum, the *mean* and *constant-scalar* combinations have very low capacity, having zero and one learnable parameters, respectively. On the other end, the *full* combinations can represent rich combination functions, taking advantage of the information in the full output distributions. The *entropy* combinations are motivated by the fact that we expect output distribution entropies to be informative to the combination function; intuitively, knowing how certain each model is should be helpful when deciding which model to give more weight to. Additionally, token-wise versions of each method further increase the capacity of the combination function. This setup allows us to study how important combination function capacity is for the performance of the adapted model, as well as how this relates to the amount of data used for learning the combination.

These combination functions can be learned without any access to the LMs' weights or internal states, and require only a forward pass through the small set used to train the combination network. We refer to the process of training the small network that parametrizes the combination function as fitting the combination function. Once the combination function is fit, the combined model outputs valid probability distributions over continuations, and can be used as a regular LM.

## 3 Experimental setup

### 3.1 Models

We use OPT-30B and OPT-1.3B (Zhang et al., 2022) as our large black-box and small white-box LMs, respectively. Our choice of OPT is motivated by the following reasons:

1. Both the small and large models must share the tokenizer in our current formulation.[1] Since we want to train the small domain experts by fine-tuning an existing model, we need a model family that has both large and small models sharing the same tokenizer, which OPT provides.

2. To rigorously determine what constitutes a new domain for the models, we need to know what data they were trained on, which is not public for most proprietary models behind APIs.[2]

We report results for the large model and the small fine-tuned model, which can be taken as the **baselines**, as well as their combination through our proposed method. For the parametrization of the combination functions, we use small neural networks, with the following architectures:

- **Constant-scalar:** A single neuron with no input, passed through a sigmoid to force it into $(0, 1)$.

- **Constant-vector:** A vector of neurons with no input, passed through a sigmoid to force it into $(0, 1)^{|V|}$.

- **Entropy-scalar:** Input layer is two-dimensional, consisting of both entropies, followed by 1D BatchNorm, two hidden layers of dimension 512, with ReLU non-linearities, and a one-dimensional output layer with a sigmoid non-linearity, to force it into $(0, 1)$.

- **Entropy-vector:** Input layer is same as for *entropy-scalar*, followed by 1D BatchNorm, two hidden layers of dimension 512, with ReLU non-linearities, and a $|V|$-dimensional output layer with a sigmoid non-linearity, to force it into $(0, 1)^{|V|}$.

- **Full-scalar:** Input layer is $2|V|$-dimensional, consisting on the concatenated output distributions for each model, followed by 1D BatchNorm, two hidden layers of dimension 512, with ReLU non-linearities, and a one-dimensional output layer with a sigmoid non-linearity, to force it into $(0, 1)$.

- **Full-vector:** Input layer same as for *full-scalar*, $2|V|$-dimensional, followed by 1D BatchNorm, two hidden layers of dimension 512, with ReLU non-linearities, and a $|V|$-dimensional output layer with a sigmoid non-linearity, to force it into $(0, 1)^{|V|}$.

We train all combination networks using the Adam optimizer and a learning rate of $2e-3$ with the exception of *constant-vector*, for which we use a learning rate of $1e-2$, and a batch size of 1024. We run optimization for a single epoch in all cases, as we found this to be enough in preliminary experiments.

Note that the **mean** combination function has no learnable parameters. Finally, we also report **max-prob oracle** results as the upper-bound, which simulates a perfect combination function that gives 100% of the weight to the best model for any given token.

## 3.2 Evaluation

For evaluation, we adapt our model for three new domains and a downstream task. The three new **domains** are defined by three datasets:

- The **Amazon Reviews** dataset (McAuley et al., 2015; He and McAuley, 2016), consisting of a large collection of reviews and ratings entered by users on the Amazon website.

- The **Enron Emails** dataset (Klimt and Yang, 2004), consisting of internal emails made public by the Federal Energy Regulatory Commission during the investigation of the Enron company.

- The **FreeLaw** subset of The Pile (Gao et al., 2021), consisting of a large collection of court opinions from federal and state courts.

For each dataset, we extract two sets of 1000 1024-token sequences, which we call *train-fit* and *test*, respectively, and use the rest for the train set. The *train-fit* sets are used to fit the combination functions, and we report perplexity on the *test* sets for evaluation. We use the train set to fine-tune OPT-1.3B using the Adam optimizer, a 1024-token sequence length, a fixed learning rate of $4e-4$, and a batch size of $1024 * 90 = 92160$ tokens. In the

---

[1]Although it is possible to either adapt LMs to a new vocabulary or extend our approach to work with different tokenizers, that would add a new dimension to our experiments, separate from the core research question that we want to study.

[2]While this is not a problem for applying our method in practice, it does rule out proprietary black-box models for scientific study.

| | Amazon | Enron | Freelaw |
|---|---|---|---|
| **OPT-1.3B FT** | 17.00 | 3.30 | 4.98 |
| **OPT-30B** | 20.37 | 5.53 | 6.50 |
| **Mean** | 15.88 | 3.47 | 4.92 |
| **Constant-scalar** | 15.80 | 3.27 | 4.84 |
| **Constant-vector** | 15.62 | 3.31 | 4.82 |
| **Entropy-scalar** | 15.50 | **3.24** | 4.78 |
| **Entropy-vector** | 15.41 | 3.24 | **4.76** |
| **Full-scalar** | **15.36** | 3.27 | 4.79 |
| **Full-vector** | 15.43 | 3.27 | 4.79 |
| **Max-prob (oracle)** | 12.59 | 2.89 | 4.12 |

Table 1: **Domain adaptation results (perplexity).** By combining a small domain expert and large general model, we achieve better perplexities than either of the original models.

case of Enron Emails we fine-tuned for a single epoch, corresponding to 3000k steps. For Amazon Reviews and FreeLaw we performed 30k steps, and had to stop well before reaching the first epoch, due to compute constraints. Unless otherwise stated, the full *train-fit* sets are used to fit the combination functions.

For **downstream evaluation**, we experiment on English-Czech and English-German MT using the WMT21 dataset (Barrault et al., 2020). We create a training set by verbalizing all the sentence pairs and concatenating them into a single corpus. Details of the verbalization templates can be found in Appendix A. We create a validation set following the same procedure on the WMT20 test set (Akhbardeh et al., 2021), and extract a *train-fit* set of 1000 1024-token sequences for fitting the combination functions, as we do in domain adaptation. Following the recommended practice in the area (Freitag et al., 2022), we use BLEURT (Sellam et al., 2020) on the WMT21 test set as our evaluation metric, and report additional results with BLEU (Papineni et al., 2002) in Appendix B. We used 3-shot prompting for evaluation, as longer sequence lenghts resulted in OOM issues in our hardware. We use the training set to fine-tune OPT-1.3B using the exact same settings described above. We train for 2k steps, corresponding to a total of around 2.5 million parallel sentences.[3]

---

[3]Although the full combined training set for English-German and English-Czech is bigger than 2.5M parallel sentences, we were interested in simulating the setting where limited translation data is available. Given enough parallel data, one can train a strong translation system from scratch, without having to adapt a generalist model.

| | en-de | en-cs | de-en | cs-en | avg |
|---|---|---|---|---|---|
| **OPT-1.3B FT** | 52.36 | 32.66 | 67.95 | 60.47 | 53.36 |
| **OPT-30B** | 54.77 | 29.21 | 68.45 | 61.83 | 53.56 |
| **Mean** | 57.62 | 35.34 | **69.84** | 63.62 | 56.61 |
| **Constant-scalar** | 57.73 | 35.08 | 69.70 | 63.70 | 56.56 |
| **Constant-vector** | 57.71 | 34.69 | 69.60 | 63.64 | 56.41 |
| **Entropy-scalar** | 57.87 | 35.18 | 69.59 | 63.88 | 56.63 |
| **Entropy-vector** | **58.11** | **35.41** | 69.44 | **64.06** | **56.76** |
| **Full-scalar** | 57.98 | 35.06 | 69.57 | 63.59 | 56.55 |
| **Full-vectors** | 58.02 | 35.31 | 69.66 | 63.37 | 56.59 |

Table 2: **MT results (BLEURT).** The learned combinations significantly outperforms both models in a downstream task, often by a substantial margin.

## 4 Results

We next present our main results on domain adaptation (§4.1) and MT (§4.2).

### 4.1 Domain adaptation

We report domain adaptation results in Table 1. We observe that the combined models are able to achieve substantially lower perplexities than either of the individual models. Even simple averaging works remarkably well, improving over both baselines in Amazon Reviews and FreeLaw, but learned combinations perform best. The *entropy-scalar* combination works best across the board, achieving a relative improvement in perplexity of 9% in Amazon Reviews, 2% in Enron Emails and 4% in FreeLaw over the best single model. This supports our hypothesis that output distribution entropies are informative to the combination function. However, higher capacity combination functions like *full-scalar* work better in some cases, as is the case for Amazon Reviews.

Overall, our results show that the adapted model is able to leverage domain-specific knowledge in the small model, as well as the knowledge in the large generalist model, in order to improve over either of them. However, there is still a significant gap between the adapted models and the max-prob oracle, suggesting gains could still be made through a better combination function.

### 4.2 Machine translation

Table 2 reports downstream results on MT. As for domain adaptation, all the learned combinations outperform both the small fine-tuned model and the large black-box model. This shows that our approach can work for adaptation to downstream tasks, and is not limited to domain adaptation. Once again, the simple *mean* combination per-

forms very well, obtaining the second best results after *entropy-vector*. In any case, the combination function has a relatively small impact in MT, and even the worst performing approach brings large improvements over the baseline.

## 5 Analysis

In this section, we study the following aspects of our approach:

- How dependent is the quality of the resulting model on the amount of data used to fit the combination function?

- How dependent is the quality of the resulting model on the amount of data used to fine-tune the small LM?

- How much is general language modeling performance degraded by domain adaptation?

- Is the learned combination interpretable?

### 5.1 Effect of the amount of data for fitting

In order to study how the performance of the adapted model varies with respect to the amount of data used to fit the combination function, we fit each combination function three times, on a varying number of tokens. We report results for the Amazon Reviews dataset in Table 3, and additional results in Appendix B.

As expected, performance improves with more training data. However, the difference varies across methods. For example, *constant-scalar*, which has a very low capacity, performs equally well when trained on 100 or 1000 sequences. On the other hand, the *full-scalar* and *full-vector* functions, that take the entire probability distribution as input, benefit from more training sequences. The *entropy-scalar* combination strikes a good balance, performing well across the board, and retaining strong performance when fit on as little as 100 sequences.

### 5.2 Effect of fine-tuning steps

Figure 2 shows the performance of the adapted models, when fine-tuning the small model for a varying number of sequences. At step 0 (i.e., before fine-tuning begins), the small LM corresponds to vanilla OPT-1.3B, which performs considerably worse than OPT-30B on Amazon Reviews. Even in that case, *entropy-scalar* performs on par with OPT-30B, while *mean* is slightly worse. This shows that learnable combination functions are able to

|  | 100 | 500 | 1000 |
|---|---|---|---|
| **OPT-1.3B FT** | 17.00 | 17.00 | 17.00 |
| **OPT-30B** | 20.37 | 20.37 | 20.37 |
| **Mean** | 15.88 | 15.88 | 15.88 |
| **Constant-scalar** | 15.80 | 15.80 | 15.80 |
| **Constant-vector** | 15.80 | 15.66 | 15.62 |
| **Entropy-scalar** | 15.51 | 15.50 | 15.50 |
| **Entropy-vector** | 15.52 | 15.45 | 15.41 |
| **Full-scalar** | 15.63 | 15.40 | 15.36 |
| **Full-vector** | 15.71 | 15.49 | 15.43 |

Table 3: **Perplexity on Amazon Reviews**, using a different number of sequences to fit the combination function. Perplexity improves with the number of sequences, but results are already strong with only 100 sequences.

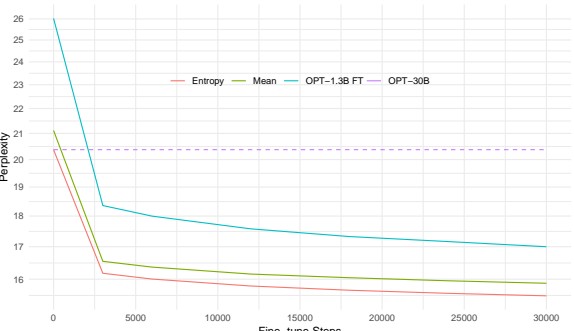

Figure 2: **Perplexity on Amazon Reviews**, varying the amount of fine-tuning steps.

avoid any loss in performance when combining with a poor domain expert. At the same time, it is also remarkable that the combination of vanilla OPT-1.3B and OPT-30B is not better than OPT-30B alone. This can also be seen in Table 4, which compares using vanilla OPT-1.3B and fine-tuned OPT-1.3B as the small model. This shows that our reported improvements do not solely come from an ensembling effect, and our proposed approach effectively combines the power of the large LM and the domain expertise of the small LM.

In addition, we observe that our combined LM substantially improves upon each individual LM as early as step 3000. In fact, the gap between the small fine-tuned LM and our combined LM slightly narrows as training progresses. For instance, for *entropy-scalar*, the gap between the small LM and the combined LM is 2.18 perplexity points at step 3000 (12% relative improvement), which goes down to 1.5 for the fully fine-tuned model (9% relative improvement). This is intuitive, as the more data is available in the target domain, the less useful will be integrating the general knowledge in the large LM.

|  | Orig | FT |
|---|---|---|
| **OPT-1.3B** | 26.03 | 17.00 |
| **OPT-30B** | 20.37 | 20.37 |
| **Mean** | 21.12 | 15.88 |
| **Constant-scalar** | 20.28 | 15.80 |
| **Constant-vector** | 20.55 | 15.62 |
| **Entropy-scalar** | 20.37 | 15.51 |
| **Entropy-vector** | 20.30 | 15.44 |
| **Full-scalar** | 20.26 | 15.41 |
| **Full-vector** | 20.30 | 15.48 |

Table 4: **Perplexity on Amazon Reviews**, using either original OPT-1.3B or fine-tuned OPT-1.3B as the small LM. The combination methods barely improve upon OPT-30B when using the former, showing that our approach does not only work due to an ensembling effect.

|  | Amazon-fit | | Mixin-fit | |
|---|---|---|---|---|
|  | **Amazon** | **Pile** | **Amazon** | **Pile** |
| **OPT-1.3B FT** | 17.00 | 19.78 | 17.00 | 19.78 |
| **OPT-30B** | 20.37 | 6.82 | 20.37 | 6.82 |
| **Mean** | 15.88 | 7.72 | 15.88 | 7.72 |
| **Constant-scalar** | 15.80 | 8.35 | 16.52 | 7.08 |
| **Constant-vector** | 15.62 | 8.38 | 15.89 | 7.18 |
| **Entropy-scalar** | 15.50 | 7.35 | 15.80 | 6.94 |
| **Entropy-vector** | 15.41 | 8.53 | 15.61 | 6.92 |
| **Full-scalar** | 15.36 | 9.31 | 15.45 | 6.85 |
| **Full-vector** | 15.43 | 10.07 | 15.48 | 6.91 |

Table 5: **Perplexity on Amazon Reviews and The Pile**, using either the former to fit the combination function (amazon-fit), or the concatenation of the two (mixin-fit).

## 5.3 Effect on general language modeling

We are also interested in measuring how well the adapted models retain the general language modeling ability of the original large model. We use perplexity on **The Pile** (Gao et al., 2021) as a proxy of general language modeling performance, as it is a large collection of many datasets from different domains, often used to train generalist LMs (Black et al., 2022; Biderman et al., 2023). To this end, we also extract random *train-fit* and *test* subsets from The Pile. While some subsets of The Pile are also present in the training data for OPT, we do not measure performance on The Pile as a benchmark for model quality, and are only interested in it as a proxy for degradation in general language modeling ability of the adapted models.

We compare fitting the combination function on the target domain *train-fit*, as done throughout the paper, as well as on the combination of the target domain and The Pile *train-fit* sets. Table 5 reports results for Amazon Reviews, and full results can be found in Appendix B.

When fitting the combination function on Amazon Reviews, we observe a significant degradation on The Pile. However, different combination methods behave differently in this regard. For example, *entropy-scalar* and *full-vector* perform similarly in Amazon Reviews (15.50 vs 15.43), but the former performs much better on The Pile (7.35 vs 10.07). It is also remarkable that The Pile perplexity of the combined model remains far better than the small fine-tuned LM (e.g. 7.35 for *entropy-scalar* vs 19.78 for the small LM), while also performing better in-domain.

When fitting the combination function on the mixin set, we observe that performance on The

Pile is almost entirely preserved, at the expense of a slight degradation on Amazon Reviews. For example, for *full-scalar*, the combination fit on the mixin set achieves a perplexity of 15.45 on Amazon Reviews and 6.85 on The Pile, both within 0.1 of the best results for each dataset.

Overall, these results show that a large model can be adapted to a particular domain while mitigating degradation in the general domain by mixing in-domain and general text to train the combination function. Additionally, we find that different combination methods exhibit different behavior when it comes to general performance degradation, even when they exhibit similar in-domain performance.

## 5.4 Is the model combination interpretable?

We next analyze whether the weights given to each model by the combination function are interpretable. Figure 3 illustrates this over a random sample from Amazon Reviews: we show which tokens are better predicted by each model, along with which model is given a higher weight for each token. Although we do not identify a clear pattern for which tokens are better predicted by each model, we do observe that the coloring in the top and the bottom visualizations match quite closely. This means that the learned combination function is quite good at predicting when each model should be given a higher weight.[4]

In order to quantitatively analyze this, we measure the Spearman correlation between the weight given by the combination function, and the actual difference in log probabilities for each token. Re-

---

[4] A perfect combination function (corresponding to the max-prob oracle in Table 1) would always give 100% of the weight to the best model for any given token, and both images would match up perfectly.

|         |                | Domain | Pile |
|---------|----------------|--------|------|
| Amazon  | **Entropy-scalar** | 0.59 | 0.71 |
|         | **Full-scalar**    | 0.44 | 0.32 |
| Freelaw | **Entropy-scalar** | 0.49 | 0.75 |
|         | **Full-scalar**    | 0.33 | 0.32 |
| Enron   | **Entropy-scalar** | 0.54 | 0.75 |
|         | **Full-scalar**    | 0.25 | 0.30 |

Table 6: **Spearman correlation between the log-probability difference of the LMs and the weight given by combination function.** Larger values mean that the learned combination is closer to the ideal oracle weighting. Rows represent adapted models on different domains and combination functions, fit on the in-domain *train-fit*.

sults are shown in Table 6. We limit our analysis to *entropy-scalar* and *full-scalar*, as they are the only ones that output a single combination factor that depends on the input. We observe significant correlations for all datasets, with *entropy-scalar* achieving better correlation than *full-scalar*, especially on The Pile. This is consistent with the results in Table 5, where *full-scalar* suffers a bigger performance loss on The Pile. Somewhat surprisingly, correlation for *entropy-scalar* is better on The Pile than on the in-domain dataset, even though the combination function is fit on the in-domain *train-fit*. One possible explanation is that The Pile better represents the training distribution of the large LM, making it better calibrated on it, which makes it easier for *entropy-scalar* to make predictions.

## 6   Related work

We present related work on domain adaptation of LMs (§6.1), and language modeling through domain experts (§6.2).

### 6.1   Domain adaptation of LMs

Domain adaptation of LMs is an extensively studied line of research. Traditional approaches include fine-tuning the model on domain-specific corpora, (Devlin et al., 2019; Liu et al., 2019; Gururangan et al., 2020), data selection on the original general corpus (Aharoni and Goldberg, 2020; van der Wees et al., 2017), and adapting or extending the tokenizer to achieve better performance on the target domain (Sachidananda et al., 2021).

Although effective, these full fine-tuning techniques are often infeasible at scale due to the excessive compute required. Some approaches aim to reduce the resources required to fine-tune large

models through parameter-efficient adaptation techniques, such as adapters (Houlsby et al., 2019), soft-prompt tuning (Liu et al., 2022), or low-rank adaptation (Hu et al., 2022). However, all of these techniques require white-box access to the original model and full backward passes, making them incompatible with black-box models.

In contrast, discrete prompt tuning approaches allow for treating the large model as a black-box (Shin et al., 2020; Sun et al., 2022; Zhang et al., 2023; Cheng et al., 2023). However, these approaches have only been proven in the limited setting of retrieving zero- or few-shot prompts that improve performance in a set of NLP tasks that the base black-box is already capable of performing, as opposed to a general method of black-box model adaptation.

Concurrent to our work, Huang et al. (2023) propose leveraging KNN retrieval from a data-store to augment an existing black-box LM. However, they only experiment with small GPT2 models as the black-box, and the adaptation depends on finding an adequate datastore, limiting application to downstream tasks such as MT.

### 6.2   Domain experts for language modeling

Another line of research explores language modeling through a combination of separate domain experts. Li et al. (2022) achieve better performance than compute-matched single transformer models and highly parallel pre-training, by training independent domain experts, and combining them at the parameter level at inference time. Gururangan et al. (2023) extend this approach to automatically discovered domain clusters. Other approaches replace components of the transformer network with independent domain-dependent modules, as is the case of Gururangan et al. (2022) for metadata-defined domains, or Pfeiffer et al. (2022) for per-language modules. All of these are pre-training approaches and seek to train better or more efficient LMs, but cannot leverage existing powerful black-box models. Our work, in contrast, seeks to adapt an existing powerful black-box through leveraging a much smaller domain expert.

## 7   Conclusions

In this work, we present a method for adapting black-box LMs to new domains and tasks, requiring access to probability-level outputs only. We first fine-tune a small domain expert white-box LM

I have never gotten tired of that cd how ever many times I listen to it I just want to listen to it again.It looks like a handkerchief hem, but it is not. More straight with a slit on each side. Just was not what I was hoping for!This has a very nice selection of cards to choose from and is very easy to use. I love putting personalized names on my cards and this lets me do thatWorks like a charm!Everybody loves Dumbo for all the right reasons - great story with humor and pathos, wonderful music, and delightful animation. However, no one seems to have noticed the underlying racial themes that fuel the plot. Dumbo's mom, and the other female elephants she lives and works with, are all Indian elephants (small ears). Dumbo's dad (Jumbo), from whom he must have inherited his big ears, must have been African. Dumbo (and his mother) were mocked and ultimately ostracized from decent elephant society because he was the product of a mixed marriage. Only after he learns (with the help of those zoot-suited, jive-talkin' crows) to use his physical "defect" to excel at something (flying) is he accepted back into the circus. While "Dumbo" teaches us that we're all "special," it also paints a rather darker picture of society being intolerant of differences unless or until those differences can benefit that society

(a) **Log-probability difference between the small and large LM.** The small fine-tuned LM gave higher probabilities to the green tokens, while the large black-box LM gave higher probability to the red ones.

I have never gotten tired of that cd how ever many times I listen to it I just want to listen to it again.It looks like a handkerchief hem, but it is not. More straight with a slit on each side. Just was not what I was hoping for!This has a very nice selection of cards to choose from and is very easy to use. I love putting personalized names on my cards and this lets me do thatWorks like a charm!Everybody loves Dumbo for all the right reasons - great story with humor and pathos, wonderful music, and delightful animation. However, no one seems to have noticed the underlying racial themes that fuel the plot. Dumbo's mom, and the other female elephants she lives and works with, are all Indian elephants (small ears). Dumbo's dad (Jumbo), from whom he must have inherited his big ears, must have been African. Dumbo (and his mother) were mocked and ultimately ostracized from decent elephant society because he was the product of a mixed marriage. Only after he learns (with the help of those zoot-suited, jive-talkin' crows) to use his physical "defect" to excel at something (flying) is he accepted back into the circus. While "Dumbo" teaches us that we're all "special," it also paints a rather darker picture of society being intolerant of differences unless or until those differences can benefit that society

(b) **Weight given to each model by *entropy-scalar*.** Tokens for which a higher weight was assigned to the small fine-tuned LM are shown in green, while tokens for which the large black-box was given a higher weight are shown in red.

Figure 3: **Difference between the small fine-tuned LM and the large black-box LM according to log-probability (a) and predicted weight (b).** The closer the two match, the better the learned combination is at predicting which model will be "right" for a given token. The text sample was chosen randomly from the Amazon Reviews testset.

on a domain-specific corpus, and then combine it with the large black-box through a combination function learned on a small fitting set, yielding an adapted LM. Additionally, our method requires only access to probability level outputs, and thus allows to leverage powerful models optimized for inference or behind APIs, without the need for white-box access to the weights. We experiment on several datasets and a downstream task, as well as performing extensive analysis of our method, reaching several conclusions:

- By combining a small domain expert and a large black-box model, the combined model outperforms either of the original ones in all cases, by as much as 9% perplexity for domain adaptation, and 6% BLEURT for MT, showing the effectiveness of our approach.

- While higher capacity combination functions can perform better when more data is used to learn the combination, lower capacity combination methods remain competitive, and perform better when learned on little data. In particular, the entropy based combinations, *entropy-scalar* and *entropy-vector*, perform well across the board, even when fit on as little as 100 sequences.

- Our approach is effective even when little

is data available to fine-tune the domain expert. In fact, the gains are biggest in this scenario, as the advantage of leveraging a good black-box generalist decreases when a big in-domain corpus is available.

- While adaptation to new domains incurs a loss of general language modeling ability, this varies per combination method, and seems to be largely mitigated by augmenting the small set on which the combination function is fit.

While our approach is effective, observed performance is still not close to the max prob oracle, which represents the ideal system where 100% of the weight is given to the best model at each time step. In future work, we would like to investigate the reasons for this gap, and potential ways of addressing it.

## Limitations

While our method requires no access to the black-box model's weights and intermediate activations, it does assume access to the full output probability distribution, which might not be available for some models served through APIs. The OpenAI API, for example, only returns the probabilities for the top 5 tokens. This is not an issue for the Constant combinations, and the Entropy methods could potentially

also be adapted, by estimating the entropy from top K probabilities.

Additionally, even though we don't fine-tune the black-box at all, our method does require a forward pass of the large black-box through a fitting set, which might potentially be costly if done through APIs, depending on pricing and the size of the fitting set.

## Acknowledgements

Aitor and Eneko were partially supported by the Basque Government (IXA excellence research group IT-1805-22; IKER-GAITU project). Aitor was supported by a doctoral grant from the Spanish MECD.

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

## A  MT verbalizations

We verbalize the MT task by first adding a prompt describing the task, and then adding several translation examples. We chunk the translation examples in blocks of 5, that is, adding 5 translation examples per verbalization. We use two different task descriptiopns, shown in Table 7, and alternate evenly between both variations to create the verbalized training corpus. For inference, we use verbalization #1 and draw 3 random translation pairs from the WMT21 development set to construct a 3-shot prompt. We draw the random translation pairs once, and keep the 3-shot prompt fixed for all models.

## B  Full results

Full results for all combination methods, dataset sizes, and evaluation settings are shown in Table 8. Table 9 reports additional MT results using BLEU.

| Verbalization #1 | Verbalization #2 |
|---|---|
| Translate the following sentences from $L1 to $L2: | Given a sentence in $L1, translate it to $L2: |
| $L1: $S1 | $L1: $S1 |
| $L2: $T1 | $L2: $T1 |
| $L1: $S2 | $L1: $S2 |
| $L2: $T2 | $L2: $T2 |
| $L1: $S3 | $L1: $S3 |
| $L2: $T3 | $L2: $T3 |
| $L1: $S4 | $L1: $S4 |
| $L2: $T4 | $L2: $T4 |
| $L1: $S5 | $L1: $S5 |
| $L2: $T5 | $L2: $T5 |

Table 7: Both verbalizations used for MT. $L1 $L2 represent the source and target languages, and $Si and $Ti represent the source and target sentences for the $i$th pair in the verbalization.

COMBINATION FUNTION FIT ON TARGET DOMAIN *train-fit*

| Dataset | Amazon Reviews | | | | | | Enron Emails | | | | | | Freelaw | | | | | |
|---|---|---|---|---|---|---|---|---|---|---|---|---|---|---|---|---|---|---|
| #Fit sequences | 100 | | 500 | | 1000 | | 100 | | 500 | | 1000 | | 100 | | 500 | | 1000 | |
| Eval domain | Dom. | Pil. | Dom. | Pil. | Dom. | Pil. | Dom. | Pil. | Dom. | Pil. | Dom. | Pil. | Dom. | Pil. | Dom. | Pil. | Dom. | Pil. |
| Mean | 15.88 | 7.72 | 15.88 | 7.72 | 15.88 | 7.72 | 3.47 | 7.45 | 3.47 | 7.45 | 3.47 | 7.45 | 4.92 | 7.56 | 4.92 | 7.56 | 4.92 | 7.56 |
| Constant-scalar | 15.80 | 8.35 | 15.80 | 8.35 | 15.80 | 8.35 | 3.27 | 9.89 | 3.27 | 9.75 | 3.27 | 9.63 | 4.84 | 8.98 | 4.84 | 8.90 | 4.84 | 8.90 |
| Constant-vector | 15.80 | 7.82 | 15.66 | 8.12 | 15.62 | 8.38 | 3.42 | 7.59 | 3.34 | 8.03 | 3.31 | 8.39 | 4.90 | 7.68 | 4.84 | 8.05 | 4.82 | 8.37 |
| Entropy-scalar | 15.51 | 7.30 | 15.50 | 7.51 | 15.50 | 7.35 | 3.24 | 8.30 | 3.24 | 8.10 | 3.24 | 8.02 | 4.78 | 7.84 | 4.78 | 8.12 | 4.78 | 8.33 |
| Entropy-vector | 15.52 | 8.10 | 15.45 | 8.20 | 15.41 | 8.53 | 3.25 | 8.22 | 3.24 | 8.53 | 3.24 | 8.18 | 4.80 | 8.05 | 4.77 | 8.05 | 4.76 | 8.05 |
| Full-scalar | 15.63 | 7.97 | 15.40 | 9.28 | 15.36 | 9.31 | 3.32 | 8.22 | 3.27 | 10.11 | 3.27 | 9.86 | 4.82 | 8.13 | 4.79 | 9.48 | 4.79 | 9.45 |
| Full-vector | 15.71 | 7.90 | 15.49 | 9.62 | 15.43 | 10.07 | 3.34 | 8.11 | 3.27 | 10.54 | 3.27 | 9.82 | 4.85 | 7.90 | 4.80 | 9.30 | 4.79 | 9.27 |
| OPT-1.3B FT | 17.00 | 19.78 | 17.00 | 19.78 | 17.00 | 19.78 | 3.30 | 12.73 | 3.30 | 12.73 | 3.30 | 12.73 | 4.98 | 15.55 | 4.98 | 15.55 | 4.98 | 15.55 |
| OPT-30B | 20.37 | 6.82 | 20.37 | 6.82 | 20.37 | 6.82 | 5.53 | 6.82 | 5.53 | 6.82 | 5.53 | 6.82 | 6.50 | 6.82 | 6.50 | 6.82 | 6.50 | 6.82 |
| Max-prob (oracle) | 12.59 | 5.93 | 12.59 | 5.93 | 12.59 | 5.93 | 2.89 | 5.89 | 2.89 | 5.89 | 2.89 | 5.89 | 4.12 | 5.75 | 4.12 | 5.75 | 4.12 | 5.75 |

COMBINATION FUNTION FIT ON MIX OF IN DOMAIN AND THE PILE *train-fit*

| Dataset | Amazon Reviews | | | | | | Enron Emails | | | | | | Freelaw | | | | | |
|---|---|---|---|---|---|---|---|---|---|---|---|---|---|---|---|---|---|---|
| #Fit sequences | 200 | | 1000 | | 2000 | | 200 | | 1000 | | 2000 | | 200 | | 1000 | | 2000 | |
| Eval domain | Dom. | Pil. | Dom. | Pil. | Dom. | Pil. | Dom. | Pil. | Dom. | Pil. | Dom. | Pil. | Dom. | Pil. | Dom. | Pil. | Dom. | Pil. |
| Mean | 15.88 | 7.72 | 15.88 | 7.72 | 15.88 | 7.72 | 3.47 | 7.45 | 3.47 | 7.45 | 3.47 | 7.45 | 4.92 | 7.56 | 4.92 | 7.56 | 4.92 | 7.56 |
| Constant-scalar | 16.56 | 7.06 | 16.47 | 7.10 | 16.52 | 7.08 | 3.57 | 7.22 | 3.56 | 7.24 | 3.56 | 7.24 | 5.18 | 6.92 | 5.16 | 6.94 | 5.15 | 6.96 |
| Constant-vector | 15.85 | 7.53 | 15.85 | 7.29 | 15.89 | 7.18 | 3.45 | 7.41 | 3.44 | 7.32 | 3.45 | 7.27 | 4.93 | 7.37 | 4.95 | 7.14 | 4.96 | 7.04 |
| Entropy-scalar | 15.87 | 6.92 | 15.71 | 6.98 | 15.80 | 6.94 | 3.38 | 6.98 | 3.35 | 7.02 | 3.36 | 7.01 | 4.91 | 6.76 | 4.88 | 6.80 | 4.92 | 6.75 |
| Entropy-vector | 15.69 | 7.07 | 15.66 | 6.96 | 15.61 | 6.92 | 3.36 | 7.12 | 3.34 | 7.04 | 3.35 | 6.98 | 4.89 | 6.92 | 4.83 | 6.90 | 4.85 | 6.78 |
| Full-scalar | 15.55 | 6.91 | 15.42 | 6.90 | 15.45 | 6.85 | 3.47 | 7.10 | 3.47 | 7.06 | 3.45 | 6.99 | 4.93 | 6.78 | 4.90 | 6.72 | 4.85 | 6.77 |
| Full-vector | 15.63 | 7.16 | 15.53 | 6.98 | 15.48 | 6.91 | 3.41 | 7.37 | 3.42 | 7.17 | 3.44 | 7.05 | 4.92 | 7.05 | 4.89 | 6.90 | 4.87 | 6.80 |
| OPT-1.3B FT | 17.00 | 19.78 | 17.00 | 19.78 | 17.00 | 19.78 | 3.30 | 12.73 | 3.30 | 12.73 | 3.30 | 12.73 | 4.98 | 15.55 | 4.98 | 15.55 | 4.98 | 15.55 |
| OPT-30B | 20.37 | 6.82 | 20.37 | 6.82 | 20.37 | 6.82 | 5.53 | 6.82 | 5.53 | 6.82 | 5.53 | 6.82 | 6.50 | 6.82 | 6.50 | 6.82 | 6.50 | 6.82 |
| Max-prob (oracle) | 12.59 | 5.93 | 12.59 | 5.93 | 12.59 | 5.93 | 2.89 | 5.89 | 2.89 | 5.89 | 2.89 | 5.89 | 4.12 | 5.75 | 4.12 | 5.75 | 4.12 | 5.75 |

Table 8: Full results for all combination methods, when fit on different amount of tokens, and on different domains. Note that the #Fit sequences is doubled when fitting the combination function on a mix of in-domain and Pile data, since the same number of tokens is drawn from each.

| | en-de | | en-cs | | de-en | | cs-en | | avg | |
|---|---|---|---|---|---|---|---|---|---|---|
| | BLEURT | BLEU | BLEURT | BLEU | BLEURT | BLEU | BLEURT | BLEU | BLEURT | BLEU |
| **Mean** | 57.62 | 14.39 | 35.34 | 5.76 | 69.84 | 26.72 | 63.62 | 21.57 | 56.61 | 17.11 |
| **Constant-scalar** | 57.73 | 13.76 | 35.08 | 5.66 | 69.70 | 26.68 | 63.75 | 21.32 | 56.56 | 16.86 |
| **Constant-vector** | 57.71 | 13.88 | 34.69 | 5.28 | 69.60 | 26.65 | 63.64 | 21.41 | 56.41 | 16.80 |
| **Entropy-scalar** | 57.87 | 13.76 | 35.18 | 5.60 | 69.59 | 26.30 | 63.88 | 21.31 | 56.63 | 16.74 |
| **Entropy-vector** | 58.11 | 14.25 | 35.41 | 5.64 | 69.44 | 26.73 | 64.06 | 21.47 | 56.76 | 17.02 |
| **Full-scalar** | 57.98 | 14.22 | 35.06 | 5.52 | 69.57 | 25.86 | 63.59 | 20.50 | 56.55 | 16.52 |
| **Full-vector** | 58.02 | 14.11 | 35.31 | 5.19 | 69.66 | 26.13 | 63.37 | 20.96 | 56.59 | 16.60 |
| **OPT-1.3B FT** | 52.36 | 15.05 | 32.66 | 5.48 | 67.95 | 25.27 | 60.47 | 19.13 | 53.36 | 16.23 |
| **OPT-30B** | 54.77 | 9.64 | 29.21 | 3.13 | 68.45 | 24.08 | 61.83 | 18.49 | 53.56 | 13.84 |

Table 9: Full BLEU and BLEURT results for MT.