# OpenReview forum: "CombLM: Adapting Black-Box Language Models through Small Fine-Tuned Models"
_EMNLP/2023/Conference — EMNLP 2023 Main_

### Official Review · Reviewer_BRRy · 2023-08-06

**Soundness:** 4

**Excitement:**

4: Strong: This paper deepens the understanding of some phenomenon or lowers the barriers to an existing research direction.

**Paper Topic And Main Contributions:**

This paper externally adapts black-box large language models with a white-box smaller language model to enable better performance to new tasks and domains. The larger model is adapted via the probability distribution, where the probability distribution of the black-box model is combined with the probability distribution of the white-box model. The paper explores a wide range of combinations of probability distributions with varying capacity.

The method is evaluated via model adaptation to new domains and downstream with machine translation. Entropy-based combination methods work well across the board, even when the model is fit with very few sentences.

**Questions For The Authors:**

No questions, this work is really cool!

**Reasons To Accept:**

This work allows interpretable adaptation of black-box language models.

The proposed method thoroughly explores different combination functions, providing a great overview of of ways to externally adapt large language models.

This method is very useful and timely given the rise in black-box language models.

The paper is well written, easy to follow and reproducible.

**Reasons To Reject:**

The work is only done in English. It would be nice to see how this method adapts large models to other languages or which combination works best in other languages.

**Reproducibility:**

4: Could mostly reproduce the results, but there may be some variation because of sample variance or minor variations in their interpretation of the protocol or method.

**Reviewer Confidence:**

4: Quite sure. I tried to check the important points carefully. It's unlikely, though conceivable, that I missed something that should affect my ratings.

---

### Official Review · Reviewer_32aC · 2023-08-06

**Soundness:** 3

**Excitement:**

3: Ambivalent: It has merits (e.g., it reports state-of-the-art results, the idea is nice), but there are key weaknesses (e.g., it describes incremental work), and it can significantly benefit from another round of revision. However, I won't object to accepting it if my co-reviewers champion it.

**Justification For Ethical Concerns:**

Violation of blind review regulations, see arXiv (https://arxiv.org/pdf/2205.12213v2.pdf).

**Paper Topic And Main Contributions:**

This paper is about a method for adapting large language models to new domains and tasks, without requiring access to their weights or intermediate activations. The main contributions of this paper are:

It proposes a simple and lightweight approach that combines a small white-box language model, fine-tuned on a domain-specific corpus, with a large black-box language model, through a combination function learned on a small validation set.
It shows that the adapted model outperforms both the small fine-tuned model and the large black-box model in several domains and a downstream task (machine translation), while using much less compute than full fine-tuning.
It analyzes the effect of the amount of data used to fit the combination function and to fine-tune the small model, as well as the effect on general language modeling performance and the interpretability of the learned combination.

But, it has been public on the arXiv (https://arxiv.org/pdf/2205.12213v2.pdf), I think this is not ok for the blind review.

**Questions For The Authors:**

How do you deal with the potential bias or ethical issues that might arise from adapting black-box LMs to new domains or tasks, especially when the source of the data is not transparent or trustworthy?
How do you plan to extend your method to other types of black-box models, such as image or speech models?

**Reasons To Accept:**

It proposes a novel and simple method for adapting black-box LMs to new domains and tasks, without requiring access to their weights or intermediate activations. This is useful for leveraging the power of large LMs that are only available as black-boxes through inference APIs, or when the computational cost of fine-tuning them is prohibitive. It explores different alternatives for the combination function that merges the probability distributions from the small domain expert LM and the large black-box LM, and studies how their capacity affects the performance of the adapted model. It also shows that output distribution entropies are informative for the combination function, and that entropy-based combinations perform well across different domains and tasks.

**Reasons To Reject:**

It assumes that the large black-box LM and the small domain expert LM share the same tokenizer, which might not be the case for some models behind APIs. This could limit the applicability of the proposed method to some scenarios.
It does not provide any theoretical analysis or guarantees for the combination functions, such as their convergence, stability, or optimality. It also does not compare them with existing methods for combining probability distributions, such as mixture models or product-of-experts models.

**Reproducibility:**

4: Could mostly reproduce the results, but there may be some variation because of sample variance or minor variations in their interpretation of the protocol or method.

**Reviewer Confidence:**

5: Positive that my evaluation is correct. I read the paper very carefully and I am very familiar with related work.

---

> ### Author Rebuttal · Authors · 2023-08-29
>
> We thank the reviewer for their comments. We comment on the questions and some of the concerns below:
> - Regarding the blind review violation, the paper was uploaded to arXiv on 22 May 2023. This is before the anonymity period, which began on 23 May 2023 (Important Dates section of https://2023.emnlp.org/ ). In fact, the existence of the online version was reported to EMNLP upon submission.
> - Regarding the tokenizer issue, while it is true that sharing the same tokenizer is a requirement, as long as the tokenizer of the black-box model is known, as is the case for example for the OpenAI API models, one can train the small expert model using this tokenizer, either from scratch or adapting an existing model with a new tokenizer, so our approach can be applied.
> - Regarding the lack of theoretical analysis, we find that the reviewer’s suggestions are vague and/or not applicable to our work. For instance, the reviewer suggests that we analyze the convergence of the combination functions, but this is not mathematically well-defined. For the convergence to be defined in the usual sense, one would need a sequence of elements belonging to a metric space. Here, we have a single combination function, not a sequence, and it is not clear how one would define a relevant distance metric to turn the space of combination functions into a metric space. It is similarly unclear how the terms “optimality” or “stability” could be defined for our combination functions so they can be analyzed theoretically. We would appreciate it if the reviewer could offer more concrete feedback.
> - Regarding the comparison to products-of-experts models, while they share the aspect of using combination functions with our work, these are not adaptation, but pre-training approaches that seek to obtain better final models. They require substantially more compute to train from scratch, and  cannot leverage a pre-trained large model, making them significantly different in spirit to our approach.
> - Regarding the adaptation of our method to other modalities, such as image or speech models, as long as the model outputs a probability distribution over a vocabulary, our combination approach could be applied, no matter if this vocabulary represents textual tokens, visual codebook tokens, or any other kind. Thus we would expect our model to be easy to extend to other modalities that employ autoregressive generation over discrete tokens.
> - Regarding potential bias or ethical issues, our approach would face the same potential issues as any other other LLM or ML method, but we don’t expect it to be an issue specific to our approach. While the combined model could inherit the undesirable behaviors of either of the original models, there is no reason our approach would exacerbate such problems or create new ones. As such, we do not think that our work poses any new bias or ethical issue.

---

### Official Review · Reviewer_ngzh · 2023-08-12

**Soundness:** 3

**Excitement:**

3: Ambivalent: It has merits (e.g., it reports state-of-the-art results, the idea is nice), but there are key weaknesses (e.g., it describes incremental work), and it can significantly benefit from another round of revision. However, I won't object to accepting it if my co-reviewers champion it.

**Paper Topic And Main Contributions:**

The authors introduce a cost-effective method to adapt these large models without needing access to their internal structures or weights. The method involves fine-tuning a smaller, accessible model and integrating it with the larger model using a minor network. This technique was tested on a large language model, showing performance improvements of up to 9% using a domain-specific model that's 23 times smaller.

**Reasons To Accept:**

 This paper presents a lightweight method for adapting large black-box language models (LMs) to new domains and tasks, assuming no access to their weights or intermediate activations. The approach fine-tunes a small white-box LM and combines it with the large black-box LM at the probability level through a small network learned on a small validation set. The paper's main contributions are a method for adapting black-box LMs to new domains and tasks, requiring access to probability-level outputs only, and the validation of this approach by adapting a large LM to several domains and a downstream task, observing improved performance in all cases. This provides a new way to adapt powerful black-box models to new domains and tasks.

The authors comprehensively test their proposed methods against Domain Adaptation nuances and also Machine Translation for Amazon Reviews, Enron, and Freelaw (all three datasets have differences in linguistic characteristics - with a good combination of varied degree of language unstructuredness, which makes the results more reliable)

**Reasons To Reject:**

Assuming access to output probability distribution is one of the major caveats of the methods presented by the authors, as models served through APIs,especially large language models cannot leverage the same framework for investigation. And given the recent interest in the investigation for LLM model, the proposed framework has limited applicability.

**Reproducibility:**

2: Would be hard pressed to reproduce the results. The contribution depends on data that are simply not available outside the author's institution or consortium; not enough details are provided.

**Reviewer Confidence:**

4: Quite sure. I tried to check the important points carefully. It's unlikely, though conceivable, that I missed something that should affect my ratings.

---

> ### Author Rebuttal · Authors · 2023-08-29
>
> We thank the reviewer for the review and their comments. Regarding the issue with APIs and requiring full access to the output distribution, we would like to point out that even when access to the full output distribution is not available, some combination functions like mean and constant-scalar can be readily used for scoring. In addition, a setting where we have access to the full output distribution of an LLM but cannot easily fine-tune it is not unrealistic: a research group or company might have an optimized inference solution for an LLM allowing to run inference over the model, but not enough resources to perform fine-tuning on it. This is realistic given that the memory and overall compute requirements of fine-tuning are significantly higher than those of inference. Our method would be readily applicable to this common use case.

---

### Meta-Review · Area_Chair_aFiE · 2023-09-25

**Recommendation:** 4

**Metareview:**

Authors propose a method to efficiently adopt a small LM with the help of a larger LM by only having access to its probability outputs on a small development/validation set, without having need to the internal model weights. Results on English show the efficacy of the proposed method. From an efficiency perspective, the paper adds a lot of practical value to the literature while not solving all the issues in this space.

---

### Decision · Program_Chairs · 2023-10-07

**Decision:**

Accept-Main

**Comment:**

Authors propose a method to efficiently adopt a small LM with the help of a larger LM by only having access to its probability outputs on a small development/validation set, without having need to the internal model weights. Results on English show the efficacy of the proposed method. From an efficiency perspective, the paper adds a lot of practical value to the literature while not solving all the issues in this space.